# Eigenvalue Initialisation and Regularisation for Koopman Autoencoders

## Abstract

Regularising the parameter matrices of neural networks is ubiquitous in training deep models. Typical regularisation approaches suggest initialising weights using small random values, and to penalise weights to promote sparsity. However, these widely used techniques may be less effective in certain scenarios. Here, we study the *Koopman autoencoder* model which includes an encoder, a Koopman operator layer, and a decoder. These models have been designed and dedicated to tackle physics-related problems with interpretable dynamics and an ability to incorporate physics-related constraints. However, the majority of existing work employs standard regularisation practices. In our work, we take a step toward augmenting Koopman autoencoders with initialisation and penalty schemes *tailored* for physics-related settings. Specifically, we propose the "eigeninit" initialisation scheme that samples initial Koopman operators from specific eigenvalue distributions. In addition, we suggest the "eigenloss" penalty scheme that penalises the eigenvalues of the Koopman operator during training. We demonstrate the utility of these schemes on two synthetic data sets: a driven pendulum and flow past a cylinder; and two real-world problems: ocean surface temperatures and cyclone wind fields. We find on these datasets that eigenloss and eigeninit improves the convergence rate by up to a factor of 5, and that they reduce the cumulative long-term prediction error by up to a factor of 3. Such a finding points to the utility of incorporating similar schemes as an inductive bias in other physics-related deep learning approaches.

## 1 Introduction

Modern neural networks are often overparameterised, i.e., their number of learnable parameters is significantly larger than the number of available training samples (Allen-Zhu et al., 2019a;b). To guide optimisation through this immense parameter space, and to potentially improve performance by avoiding overfitting, neural networks are trained with regularisation techniques (Goodfellow et al., 2016). The importance of regularisation has been shown in the theory and practice of deep learning. Prominent examples include the initialisation of parameter matrices (He et al., 2015; Hanin & Rolnick, 2018), and constraining the parameters' norm via loss penalties (Hinton, 1987; Krogh & Hertz, 1991). Initialising weights and penalising them with small random values and weight decay are arguably the most common regularisation techniques employed in training deep models with stochastic gradient descent algorithms. However, specific neural architectures, data domains, and learning problems may require different initialisation and penalty schemes. In this paper, we empirically study the effect of regularisation on physics-aware architectures.

The ground-breaking success of deep learning in solving complex tasks in vision and other domains has inspired the physics community to develop deep models suited to deal with real-world problems arising in the field (Willard et al., 2020; Karniadakis et al., 2021). In this context, we focus on dynamical systems analysed and processed using *Koopman-based* approaches (Takeishi et al., 2017; Lusch et al., 2018). Koopman theory (Koopman, 1931) proves that under certain assumptions, nonlinear and finite-dimensional systems can be transformed to a linear (albeit infinite-dimensional) representation via the Koopman operator. Using finite-dimensional approximations of this Koopman operator is advantageous as they facilitate the analysis and understanding of dynamical systems by utilising linear analysis tools. Despite the theoretical and practical advances that have significantly improved Koopman-based learning methods, the majority of existing models still apply regularisation practices designed for general neural networks. Our investigation aims to answer the research

question: can one exploit properties of Koopman operators to improve regularisation and to promote better overall performance?

The Koopman operator is a linear object with a complex spectrum. Some groups in the past have saught to place a bias on the operator's form. For example, Pan & Duraisamy (2020) propose a skew-symmetric tridiagonal form to guarantee stable Koopman models. However, there has not been a systematic investigation that evaluates the effect of initialisation and penalty regularisation schemes on the behavior of Koopman-based neural networks. The overarching objective of this paper is to help bridge this gap. Key to our regularisation schemes is the observation that spectral properties of linear Koopman operators follow a *typical structure*, shared by many dynamical systems. Namely, Koopman eigenvalues of stable dynamical systems are constrained in the unit circle of the complex plane (Mauroy & Mezić, 2016).

Motivated by the theoretical observation that, for this important class of stable systems, eigenvalues need to be within the unit circle, we propose two novel regularisation techniques: random eigenvalue generation from known distributions to initialise key weights in the network ("eigeninit"), and a loss penalty on the eigenvalues to alter their expansion ("eigenloss"). Importantly, our regularisation schemes can be incorporated with all Koopman-based approaches as a means to regularise the associated Koopman operator. We evaluate our approach on several challenging physics-related datasets and in comparison to standard regularisation and initialisation approaches as well as a state-of-the-art baseline. Our results indicate that our spectral schemes for initialisation and penalty improve the performance of Koopman-based networks, leading to faster and smoother convergence in the objective loss, as well as yielding models that generalise better in long-term prediction tests. Although these schemes require tuning, a guide to which we provide, they are easily applicable in any context where a Koopman approximate is used. As such, they present a useful tool for practitioners in the field.

## 2 RELATED WORK

Regularisation of neural networks is a fundamental research topic in machine learning. Common regularisation techniques include dropout (Srivastava et al., 2014), batch normalisation (Ioffe & Szegedy, 2015), and data augmentation (Perez & Wang, 2017). In what follows, we mainly discuss parameter initialisation approaches and weight penalty methods.

**Parameter initialisation.** Proper initialisation of deep models is known to be crucial to their successful training (Sutskever et al., 2013). Standard initialisation schemes for $\tanh(\cdot)$ (Glorot & Bengio, 2010) and ReLU (He et al., 2015) activations sample small random values for the weight matrices of the network. These choices are backed by theoretical results showing that initial small scale weights generalise better (Woodworth et al., 2020), and random initialisation converges to local minimisers on differentiable losses (Lee et al., 2016). While guaranteed, convergence may be exponentially slow (Du et al., 2017). Other approaches promote information flow using initial orthogonal weight matrices (Saxe et al., 2014; Mishkin & Matas, 2016; Pennington et al., 2017). Koopman-based approaches (Pan & Duraisamy, 2020) suggest to initialise the Koopman operator using the dynamic mode decomposition (DMD) estimates (Schmid, 2010). Still, the general problem of weight initialisation in neural networks remains an active research topic in practice (Arpit et al., 2019), and theory (Hanin & Rolnick, 2018; Stöger & Soltanolkotabi, 2021).

**Parameter loss penalties.** Penalising the norms of weight matrices using $L_1$ and $L_2$ metrics is a common practice in machine learning, collectively termed weight decay (Hinton, 1987). More recently, Arjovsky et al. (2016) showed that recurrent neural networks can avoid the issue of exploding gradients if their hidden-to-hidden matrices are parameterised to be unitary. Similarly, Yoshida & Miyato (2017) introduce a regularisation scheme based on penalising the spectral norm of weight matrices. Greydanus et al. (2019) assume the underlying system is measure-preserving, and their network learns the Hamiltonian during training. Lusch et al. (2018) use block diagonal Koopman operators to support continuous spectra. To promote stability, Erichson et al. (2019) employ Lyapunov-based constraints, whereas Pan & Duraisamy (2020) guarantees that Koopman eigenvalues remain in the unit circle via tridiagonal Koopman operators. In contrast, a soft penalty on the forward and backward dynamics was introduced by Azencot et al. (2020), yielding stable Koopman systems and state of the art performance.

## 3 BACKGROUND

### 3.1 KOOPMAN OPERATOR THEORY

Suppose we have a discrete time dynamical system given by:

$$x_{k+1} = \varphi(x_k) \,, \ k \in \mathbb{N}_0 \,. \tag{1}$$

Let $f : S \to \mathbb{R}$ (where $S$ is the system's state space) be a real-valued observable of the system. By real-valued observable we mean a function of the state of the dynamical system, where an observed data-variable is its range. The collection of all such observables forms a linear vector space. The Koopman operator $K$ is a linear transformation on this functional space:

$$Kf(x) = f \circ \varphi(x) \,. \tag{2}$$

Essentially, the Koopman operator may be viewed as a *lifting* of the dynamics from the state space to the space of observables (Mezić, 2015). Whilst the Koopman operator maps between function spaces and is thus infinite dimensional, it has been shown empirically that finite-dimensional approximations ($U$) are generally quite expressive, where $U$ is usually found through dynamic mode decomposition or deep learning (Mauroy et al., 2020).

### 3.2 KOOPMAN AUTOENCODERS

Pioneered by several groups (Takeishi et al., 2017; Lusch et al., 2018), the Koopman autoencoder (KAE) is a deep neural network designed to solve physics-related problems. Primarily, KAEs are based on an autoencoder architecture (Hinton & Zemel, 1993) which has a "bottleneck" structure including an encoder $\psi$ which learns a low-dimensional representation of the input signal, and a decoder $\omega$ which recovers the original signal from the low-dimensional code. Koopman autoencoders extend these family of autoencoder models by introducing a Koopman operator module in-between $\psi$ and $\omega$ whose purpose is to produce a finite-dimensional approximation of the Koopman operator. The Koopman module $U$ is essentially a linear layer (with no bias) aiming at advancing latent codes forward in time. Put together, the KAE architecture can be described by the following equations,

$$y_k := \psi(x_k) \,, \tilde{x}_k := \omega(y_k) \,, \hat{y}_{k+1} := Uy_k \,, \hat{x}_{k+1} := \omega(\hat{y}_{k+1}) \,, \tag{3}$$

where $x_k$ is the input signal, $\tilde{x}_k$ is its reconstruction, and $\hat{x}_{k+1}$ is the reconstruction of the latent code obtained with the Koopman matrix $U$. The model is trained with a reconstruction loss $\text{MSE}(x_k, \tilde{x}_k)$, and a prediction loss $\text{MSE}(x_k, \hat{x}_{k+1})$, where MSE is the mean squared error. Several existing methods adopt this general framework which we illustrate in Fig. 1, and they further generalize it to promote stability (Pan & Duraisamy, 2020; Azencot et al., 2020), to incorporate control (Han et al., 2021), and to other settings (Morton et al., 2018; Li et al., 2020; Yeung et al., 2019; Otto & Rowley, 2019; Iwata & Kawahara, 2020).

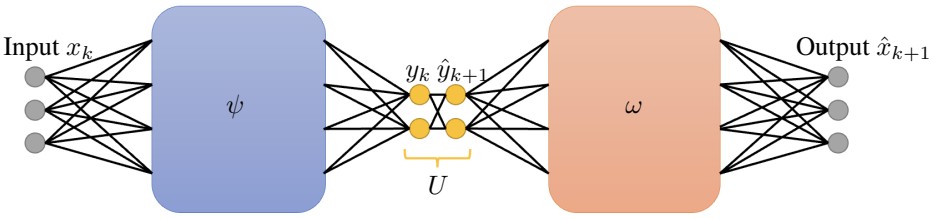

Figure 1: Diagram of Koopman autoencoder architecture.

### 3.3 SPECTRAL ANALYSIS OF KOOPMAN OPERATORS

We briefly mentioned above that one of the main advantages to working with Koopman-based approaches, and KAE in particular, is the linearity of the Koopman matrix $U$. Specifically, we can

harness tools from linear analysis and spectral approaches to study the underlying dynamics (Strogatz, 2018). For instance, the eigendecomposition of $U$ to eigenvectors and eigenvalues is instrumental to a qualitative characterisation of the behavior of the system. We call $(v_j, \lambda_j)$ an eigenvector-value pair of the Koopman matrix if they satisfy: $Uv_j = \lambda_j v_j$ where $\lambda_j \in \mathbb{C}$. The effect of the eigenvalues on the behavior of the system can be realized by considering the long-term evolution of inputs. Denote by $V$ the eigenvectors in columns, and $\Lambda$ the diagonal matrix with the eigenvalues along its main diagonal, then we can advance the latent code $y_k$ by

$$\hat{y}_{k+l} = U^l y_k = V \Lambda^l V^{-1} y_k. \tag{4}$$

Thus, advancing $y_k$ forward in time from time $k$ to time $k+l$ amounts to simply multiplying $y_k$ with powers of $U$. Eq. 4 also shows that every eigenvector $v_j$ is scaled by its associated $\lambda_j^l$, and therefore, the modulus of $|\lambda_j|$ determines its long-term behavior. We identify three qualitatively different evolution profiles, determined by the modulus of eigenvalues: if $|\lambda_j| < 1$ then the associated action of its eigenvector diminishes with time (i.e. $l \to \infty$), if $|\lambda_j| > 1$, then the action increases arbitrarily when $l \to \infty$, and if $|\lambda_j| = 1$ then the dynamics remain stable. In this way, the eigenvalues of the Koopman matrix encode the "memory" of an evolution rule.

Our regularisation schemes described in Sec. 4 are based on a prior soft bias on the number of eigenvalues in each dynamical mode – vanishing, exploding and stable.

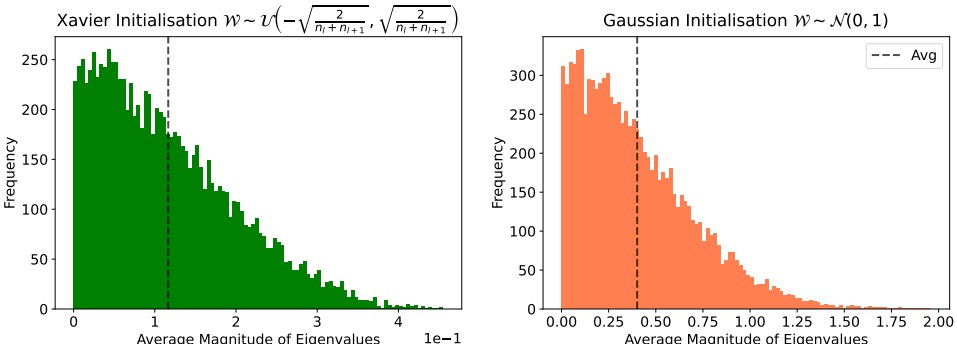

Figure 2: Eigenvalue distribution of various initialisation schemes for a $4 \times 4$ operator with six layers.

## 4 METHODS

Our discussion in Sec. 3 suggests that the optimal eigenvalue spectra of the approximate Koopman matrix may change based on the nature of the system. In our experiments we focus on the class of systems for which the eigenvalues are approximately contained in the unit circle. However, one could easily adapt our schemes to promote different distributions.

### 4.1 EIGENVALUE INITIALISATION OF LEARNABLE PARAMETERS

Initialising a deep neural network of any architecture is critical to its training and model performance. In Fig. 2 we empirically evaluate the eigenvalue distribution of parameter matrices as attained by two of the most common initialisation schemes: Xavier (Glorot & Bengio, 2010) and a random Gaussian initialisation. We find that the average magnitude of eigenvalues are approximately $0.1$ for Xavier, whilst they are $0.4$ for a Gaussian initialisation. Overall, these initialisations produce Koopman matrices whose eigenvalues span a distribution with a low variance. However, these spectra may negatively affect training in cases where the optimal Koopman matrix consists of eigenvalues with modulus $|\lambda_j| \approx 1$.

Using initial small random values for the parameter weights of deep neural networks is motivated by theoretical results which show that such initialisations converge to local minima on differentiable loss functions when solved with stochastic gradient descent (Lee et al., 2016). However, convergence may be extremely slow (Du et al., 2017). Can we improve convergence speed and model generalisation by developing initialisation schemes tailored to Koopman autoencoder architectures? Typical Koopman-based approaches utilise one of the two initialisation schemes above (see e.g., Azencot et al. (2020));

however, these choices are agnostic to the structure of the Koopman operator layer. Other techniques employ a DMD-based initialisation (see for instance, Pan & Duraisamy (2020)), which may be an unrealistic choice for highly-nonlinear and high-dimensional dynamical systems (Zhang et al., 2017). Inspired by the discussion in Sec. 3, we would like to exploit the unique role of the approximate Koopman operator $U$ as an evolution matrix and its special structure.

Therefore, we choose to focus on eigenvalue-based initialisation rather than element-wise initialisation whose effect on the eigenvalues is indirect, and leads to eigenvalues with lower magnitudes. Namely, we propose initialisation schemes that directly control the eigenvalues of the Koopman matrix $U$ and their distribution. To this end, we propose the following eigenvalue initialisation method (termed "eigeninit"): we start with an element-wise sampling from a Gaussian distribution to produce $U_0$. Then, we perform an eigendecomposition of $U_0 = V \Lambda V^{-1}$, where $V$ is a matrix with the eigenvectors $v_j$ organized in its columns, and $\Lambda$ is a diagonal matrix with the eigenvalues $\lambda_j$ along its main diagonal. Each eigenvalue $\lambda_j$ is, in general complex, and can be re-written as $r_j \, \mathrm{e}^{\mathrm{i}\theta_j}$. We modify the modulus $r_j = |\lambda_j| \in \mathbb{R}$ of the eigenvalues without altering their phases $\theta_j$. Each modulus $r_j$ is sampled from a pre-defined distribution $D$; $r_j \mapsto \tilde{r}_j \sim D$. Care is taken for complex-conjugate eigenvalue pairs: the modified modulus of both eigenvalues from a complex-conjugate eigenvalue pair is kept the same, $r_j \, \mathrm{e}^{\pm \mathrm{i}\theta_j} \mapsto \tilde{r}_j \, \mathrm{e}^{\pm \mathrm{i}\theta_j}$. Then, the initial Koopman matrix $U$ is defined via $U := V \tilde{\Lambda} V^{-1}$, where $\tilde{\Lambda}$ is a diagonal matrix with $\tilde{\lambda}_j = \tilde{r}_j \, \mathrm{e}^{\mathrm{i}\theta_j}$ along its main diagonal. Following this procedure and given that the initial matrix $U_0$ is real, then $U$ is also real (up to machine precision). All other weights in the network are initialised according to the He initialisation as we utilise ReLU activation layers (He et al., 2015).

Taking into consideration the above discussion, a natural question arises: what distributions $D$ should we sample from? If one is not looking to model divergent dynamics, the focus should be on parameter matrices $U$ initialised such that their spectral radius satisfies $\rho(U) \leq 1$. On the other hand, existing works on recurrent neural networks (Arjovsky et al., 2016) advocate the use of orthogonal weights for which $\rho(U) = 1$. In what follows, we prefer the former option, i.e., $U$ matrices with $\rho(U) \leq 1$ since orthogonal and unitary matrices are less expressive (Kerg et al., 2019). In particular, $\rho(U) \leq 1$ allows $U$ to immediately capture any dissipative dynamics in a system. In practice, we sample the eigenvalue moduli from a spike and slab distribution: $D(\text{spikeAndSlab}) := \theta \, \mathcal{U}(0,1) + (1 - \theta) \, \delta(1)$ where $\mathcal{U}(0,1)$ is a uniform distribution from 0 to 1[1] and $\delta$ is the delta function. This choice is motivated by the notion of an intrinsic dimension of the system, which will naturally give rise to a certain number of eigenvalues on the unit-circle (corresponding to conserved quantities) and others inside the unit-circle (corresponding to dissipative or finite-time horizon dynamics of the system). Whilst $\theta \in [0, 1]$ (which governs the proportion of modulus 1 eigenvalues sampled) may be chosen through a hyperparameter search, another feasible approach is to use a dynamic mode decomposition to estimate the number of non-unitary eigenvalues, and use this to inform $\theta$.

## 4.2 EIGENVALUE REGULARISATION OF LEARNABLE PARAMETERS

While proper initialisation of weights is crucial for a neural network to start optimisation from a successful trajectory that generalises well (Stöger & Soltanolkotabi, 2021), we note that without an additional mechanism that penalises non-preferable solutions, optimisation may become stuck on inferior local minima. In our setting, this means that the resulting Koopman matrix may settle on a spectrum which is not commensurate with the system being studied.

Similarly to our initialisation scheme which directly affects the eigenvalues of $U$ (Sec. 4.1), we propose a penalty method (termed "eigenloss") which biases the spectrum of the approximate Koopman operator: we augment the reconstruction and prediction loss components of the Koopman autoencoder (Sec. 3) with a new and novel loss term $\epsilon_\lambda$ whose role is to penalise eigenvalues of $U$ which are too distant from a certain value or distribution. While we could have used the same ansatz as in Sec. 4.1, promoting the spectral radius of $U$ to satisfy $\rho(U) \leq 1$, we find that on our datasets, encouraging the eigenvalues to be close or on the unit circle yields better models in terms of convergence and generalisability features. Thus, we consider the penalty scheme $\epsilon_\lambda(\text{MSE}) := \sum_j ||\lambda_j| - 1|_2^2$ where $\lambda_j$ is an eigenvalue of $U$. That is, we encourage the modulus of the eigenvalues to be approximately one. Notably, backpropagation through the eigenvalues of a matrix is **always** numerically stable, even if there are repeating eigenvalues.

---

[1] We also experimented with a uniform distribution between 0.9 and 1

The choice of eigenvalue penalty imposes different soft biases on the Koopman approximation $U$. Based on the domain-knowledge of the relevant dynamical system, one can tune the above penalties better. For instance, if the underlying system is measure-preserving, we can weight $\epsilon_\lambda$ more strongly with respect to the reconstruction and prediction losses. In contrast, if the dynamics are dissipative, we can use a weaker weight. As with initialisation, one might use the derived $\theta$ to guide this choice. If $\theta$ is near zero, it is likely better to drive eigenvalues closer to the unit circle.

## 5 EXPERIMENTS

The empirical results for eigeninit and eigenloss applied to the standard Koopman autoencoder[2] and consistent Koopman autoencoder are demonstrated in various experiments. Both were implemented in Pytorch (Paszke et al., 2019).

### 5.1 DATASETS

Datasets were chosen as a combination of synthetic and real-world data, varying in complexity and including varying amounts of dissipation. Our initial dataset was a driven frictionless pendulum governed by: $\mathrm{d}^2x/\mathrm{d}t^2 + \omega_0^2 x = f_0 \sin(\omega t)$, where $\omega_0$ the characteristic frequency, and $f_0$ and $\omega$ the amplitude and frequency of the forcing[3].

We also applied our techniques to cyclone wind fields, flow over a cylinder and sea surface temperatures. The cyclone prediction dataset was extracted from the ERA5 Re-Analysis dataset provided by ECMWF using the International Best-Tracks Archive for Climate Stewardship (IBTrACS). Prediction sequences were sub-sampled from the $u$ component of wind at the pressure level of $650\,\mathrm{hPa}$ in a $20° \times 20°$ sliding window around the cyclone. The sea surface temperature benchmark was taken from (Reynolds et al., 2007). It is a subset of the NOAA OI SST V2 High Resolution Dataset over the gulf of Mexico with a spatial resolution of $100 \times 100$ spanning a time horizon of 1305 days. Finally, the fluids dataset modeled flow over a cylinder and was produced by Kutz et al. (2016). We took the $u$-component of the flow vector field.

### 5.2 TEST AND VALIDATION PREDICTION ERROR WITH DATASETS

One can see the application of different schemes (including their combination) and the influence of these on the prediction test prediction across given prediction horizons and validation loss in Fig. 3. In addition, the state of the art consistency architecture presented by Azencot et al. (2020) is shown as an additional baseline and an architecture to which our schemes are applied. For a summary of results see Tab. 1. Of particular note is that all of our schemes (except for their combination with the pendulum dataset) outperform the standard KAE and do so by up to a factor of 3 in cumulative test error. Additionally, our initialisation schemes increased convergence rates (see Appendix C for a summary). Regarding the consistency benchmark, our schemes outperform or match its performance in all datasets except for the pendulum. In addition, augmentation of the consistent KAE with eigenloss or eigeninit improves its performance on the real-world data and rate of convergence on all datasets.

### 5.3 INVESTIGATING EIGENVALUE EVOLUTION

Although we have demonstrated the utility of our initialisation and regularisation schemes, we have yet to investigate the evolution of eigenvalues within the networks themselves. We can plot a heatmap of eigenvalue magnitudes of the Koopman operator during training (Appendix B Fig. 6). By doing so, we see that the control network has increasing eigenvalues even though there is no bias towards this and that using the unit circle MSE penalty promotes this tendency. Alternatively, with initialisation close to the unit circle, the eigenvalues tend towards the different eigenvalue classes mentioned in Sec. 3.3.

---

[2]We take the standard Koopman autoencoder to be the architecture used by Azencot et al. (2020) without the consistency modifications.

[3]We use $\omega_0 = 3.13$, $f_0 = 1$, and $\omega = 1$, and initial conditions $x \in [-\pi, \pi]$ and $\mathrm{d}x/\mathrm{d}t \in [-1, 1]$.

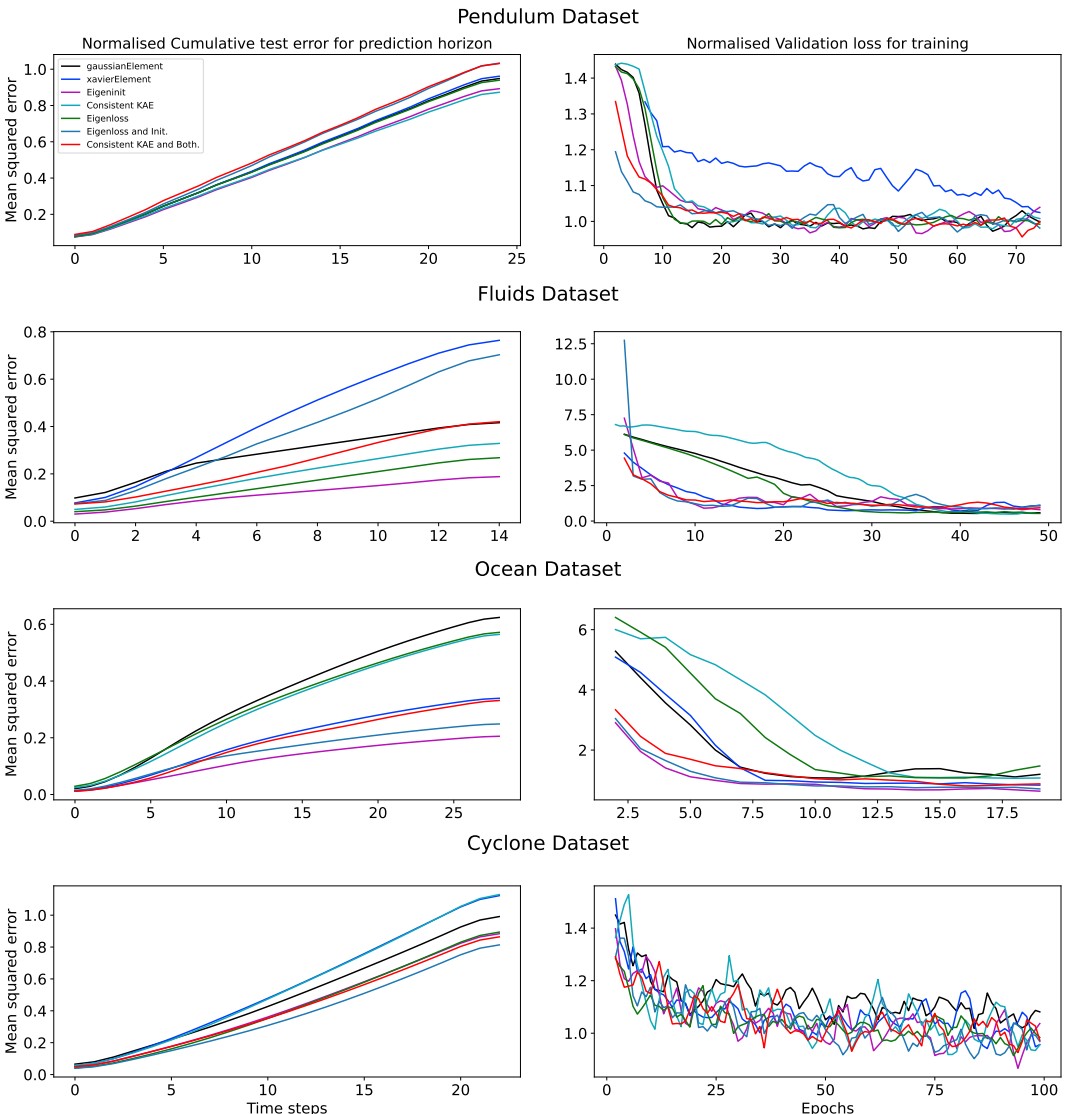

Figure 3: Averaged prediction horizon test loss and training validation loss for selected library members of eigeninit and eigenloss. Models are trained on the longest prediction horizon shown in the test loss graphs. Hyper parameters are given in appendix A.

## 5.4 COMPUTATIONAL COMPLEXITY

One reasonable concern with our regularisation technique is computational complexity. The eigen-decomposition of the Koopman operator must be applied at each iteration in order to calculate the loss with respect to the eigenvalues, which scales as $\mathcal{O}(n^{2.376})$ (Demmel et al., 2007). Thus, it is possible that our techniques do not actually converge faster, at least in terms of actual CPU wall time (as opposed to epoch number). However, this concern is rendered largely irrelevant by the size of Koopman approximations used. Even when modelling cyclone paths, which are considered a relatively complex dynamical system, we only required a Koopman matrix of size $16 \times 16$. This is an advantage of compressing the problem to a smaller latent space with the autoencoder architecture. Hence, it is likely that the computational complexity of our methods is not a concern. Indeed, as shown in Fig. 8 (Appendix E), we see that our methods still converge faster in terms of absolute wall time compared to the control.

Table 1: Minimum validation loss and final cumulative test prediction error for benchmark data sets. See bold for minimum.

| | Validation Loss | | | | | | |
|---|---|---|---|---|---|---|---|
| *Dataset* | None | Xavier | Cons. | Init | Loss | Both | Cons. & Both |
| Pendulum | **10.8** | 11.68 | **10.9** | 11.01 | 11.05 | 11.09 | 11.01 |
| Fluid | **0.19** | 0.24 | **0.18** | 0.24 | 0.29 | **0.19** | 0.26 |
| Ocean ($\times$ 10) | 0.26 | 0.21 | 0.27 | **0.14** | 0.27 | 0.16 | 0.19 |
| Cyclone | 16.18 | 14.60 | 13.77 | **14.25** | 15.44 | 14.34 | 14.33 |
| | Cumulative Test Error | | | | | | |
| Pendulum | 176.69 | 178.45 | **161.68** | 165.71 | 174.76 | 181.78 | 191.47 |
| Fluid | 1.25 | 2.30 | 1.00 | **0.58** | 0.82 | 0.96 | 1.04 |
| Ocean | 3.12 | 1.69 | 2.82 | **1.02** | 2.85 | 1.24 | 1.66 |
| Cyclone | 554.5 | 626.4 | 631.29 | 494.2 | 501.3 | **458.5** | 484.7 |

## 5.5 ABLATION WITH HYPERPARAMETERS

We also tested the sensitivity of our methods to the choice of hyperparameters on the fluids and ocean datasets. In particular, we completed ablations with differing values of $\epsilon_\lambda$ for the penalty term, $\theta$ for the spike and slab distribution, the size of the Koopman approximate and the width of the encoder and decoder. The results of this can be seen in Fig. 4. The ocean dataset always improved compared to the standard Koopman autoencoder for all hyperparameter choices. However, performance on fluids was sensitive to operator size and $\theta$.

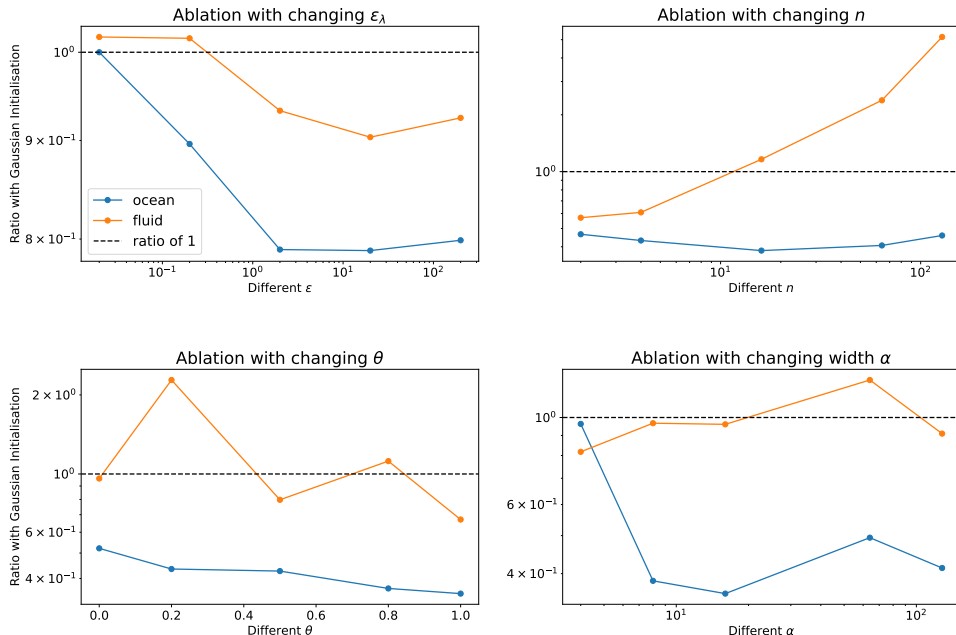

Figure 4: Testing the sensitivity of schemes to changes in network hyperparameters. Each data point is the averaged final cumulative test error. For ablation with changing width and Koopman operator size, the scheme compared is eigenloss for the fluid dataset and initialisation for the ocean dataset.

## 5.6 DYNAMIC MODE DECOMPOSITION INFORMS INITIALISATION AND REGULARISATION

The use of a spike and slab distribution naturally raises the question of how to choose the parameter $\theta \in [0, 1]$, which governs the proportion of modulus 1 eigenvalues sampled. Whilst we could do so through a simple hyperparameter search, another interesting approach could be to use a dynamic mode decomposition to give an estimate of the number of non-zero eigenvalues and hence the magnitude of $\theta$. Dynamic mode decomposition (DMD) achieves the dimensionality reduction of a

dynamical system by finding the eigenvectors associated with fixed frequency and growth (Schmid, 2010). We hypothesised that using the DMD to find an approximation of the evolution matrix $A$, with $x_{k+1} = Ax_k$, may give a computationally cheap way to estimate the number of unitary eigenvalues. We could then determine the proportion of eigenvalues to sample from the unit circle, and the proportion to sample from a zero-centred Gaussian, using this estimation. To test this, we computed the $n \times n$ DMD evolution matrix $\tilde{A}$ of the ocean dataset, matching the size of the $n \times n$ Koopman approximation in our autoencoder. We then estimated $\theta$ by the proportion of $\tilde{A}$'s eigenvalues with magnitude within 1e-3 of 1. We see from Fig. 5 that our estimated $\theta$ gave the lowest average final loss.

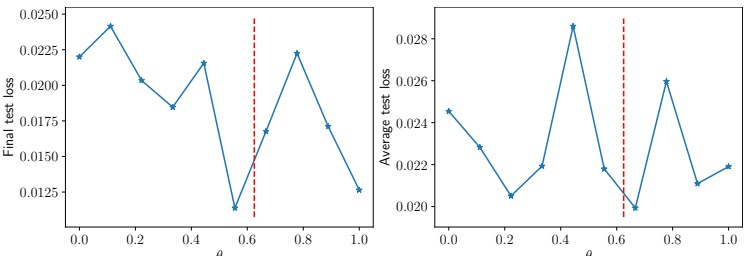

Figure 5: Examining the eigenvalues of the DMD of the ocean dataset gave the best initialisation hyperparameter $\theta$ (estimated $\theta$ shown in red).

## 6 DISCUSSION

In this research we have asked whether an initial and ongoing bias on the eigenvalues of Koopman autoencoders would improve their accuracy and convergence. We tested this via empirical analysis on the field's existing benchmarks and through the introduction of a new one (cyclone wind fields). It seems clear that certain well developed biases do in fact have utility for training these networks. In particular, our schemes outperform state of the art baseline models which do not use them. In addition, they can augment one such baseline introduced by Azencot et al. (2020), and improve its performance on real-world datasets.

These techniques are important because they marry together the two purposes of dynamical systems modelling: prediction and explanation. Whilst deep autoencoders have frequently been used for predicting complex dynamical systems, they are often criticised because they lack explainability. Koopman autoencoders are a front-running scheme to address this. However, inserting a Koopman approximation between the encoder and decoder introduces a bottleneck in the network and reduces its capacity for learning.

Because the linear dynamics are so useful in understanding the system, we want to be able to keep this interpretability whilst providing necessary forecasting capacity. The results above demonstrate that this is possible by exploiting the spectral properties of the Koopman approximation $U$. Specifically, we have shown that Koopman autoencoders using our eigeninit and eigenloss techniques *(i)* converge faster; *(ii)* reach a lower validation loss level on real-world data; *(iii)* achieve better results across multiple prediction horizons than state of the art baseline networks; and *(iv)* can productively augment other inductive biases on the Koopman approximate. Our ablations have shown the effect of various hyperparameters on model performance, depending on the nature of the dataset. Further, dynamic mode decomposition (DMD) has been shown to provide an informative way to choose the initialisation distribution. Further applications of the eigenvalues from DMD may provide insight into the conservative or dissipative aspects of the relevant system, and thus inform the appropriate eigenloss to use in regularisation.

These techniques will be of practical help for those using Koopman methods in dynamical systems. It is likely that the schemes presented here can be extended in many ways, particularly by devising better initialisation and regularisation candidates and tuning them to the nature of the problem. Whilst application to other autonomous dynamical systems is evident, it may also be possible to extend the Koopman initialisation and regularisation techniques for input-driven systems, a significant area of future potential research.

## REPRODUCIBILITY STATEMENT

In order to ensure reproducible results, we have provided an anonymised source code in *Supplemental materials* with a synthetic pendulum dataset, along with the scripts used for data processing and pipelining in this case. The Koopman autoencoder with all combinations of our techniques is included as a model, as well as other previous state-of-the-art baselines for comparison. Further, the training script used is included. Using these models and data, training runs were performed several times and averaged. Other datasets are available upon request from the referenced author. The full source code will be released upon conclusion of Open Review.

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

# A    HYPERPARAMETERS FOR MAIN RESULTS

Table 2: Hyperparameters for results in Fig. 3 and Tab. 1. Noteably the ocean and pendulum datasets had uniform distributions between 0.9 and 1 rather than between 0 and 1 as it worked slightly better.

| *Dataset* | Operator Size | Eigeninit $\theta$ | Eigenloss $\epsilon$ |
|---|---|---|---|
| Pendulum | 8 | 0 | $10^3$ |
| Fluid | 16 | 1 | 20 |
| Ocean | 16 | 0.7 | 2 |
| Cyclone | 16 | 0.7 | 20 |

# B EIGENVALUE DISTRIBUTION HEATMAPS

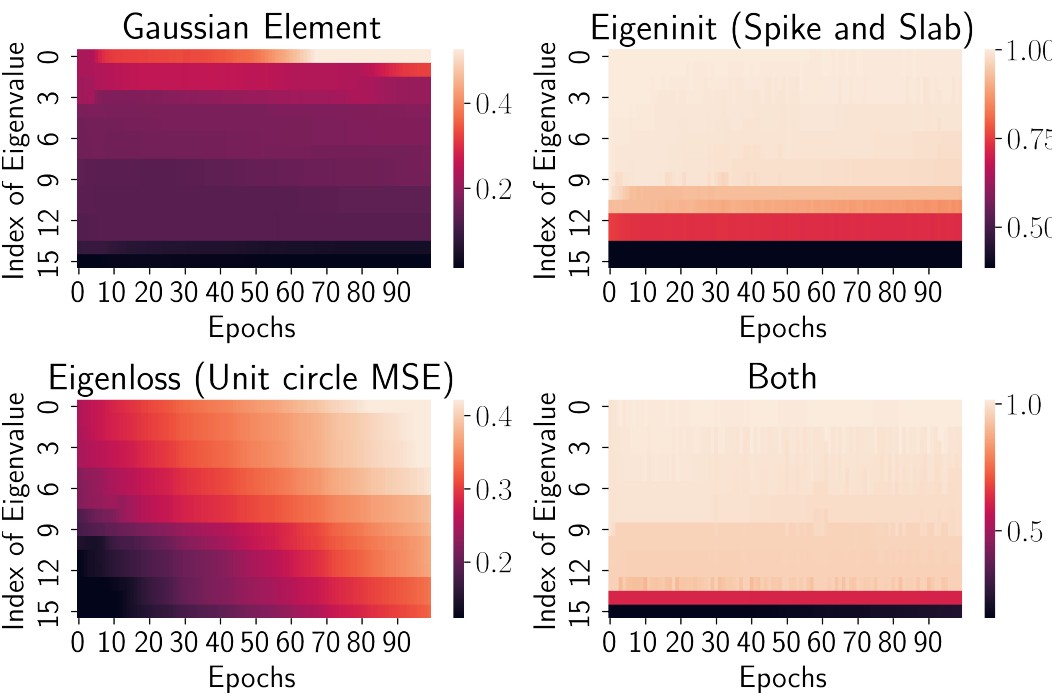

Figure 6: Change in eigenvalue magnitudes over time.

# C  CONVERGENCE STATISTICS

The convergence of solutions also improves with eigeninit and eigenloss. One can see this in Tab. 3 where the approximate epoch of convergence is noted for different data sets and scheme. Convergence was defined based on the ratio of the maximum validation difference between two points and the current validation difference between two points (excluding the first 5 epochs).

Table 3: Epoch of model convergence for benchmark data sets. See bold for minimum.

| Dataset | Regular KAE | Eigeninit | Eigenloss | Both |
|---------|-------------|-----------|-----------|------|
| Pendulum | 15 | 13 | 15 | **9** |
| Fluid | 42 | **9** | 33 | 11 |
| Ocean | 13 | **5** | 14 | 18 |
| Cyclone | 10 | **5** | 11 | 6 |

# D  WEIGHT DECAY INEFFECTIVENESS

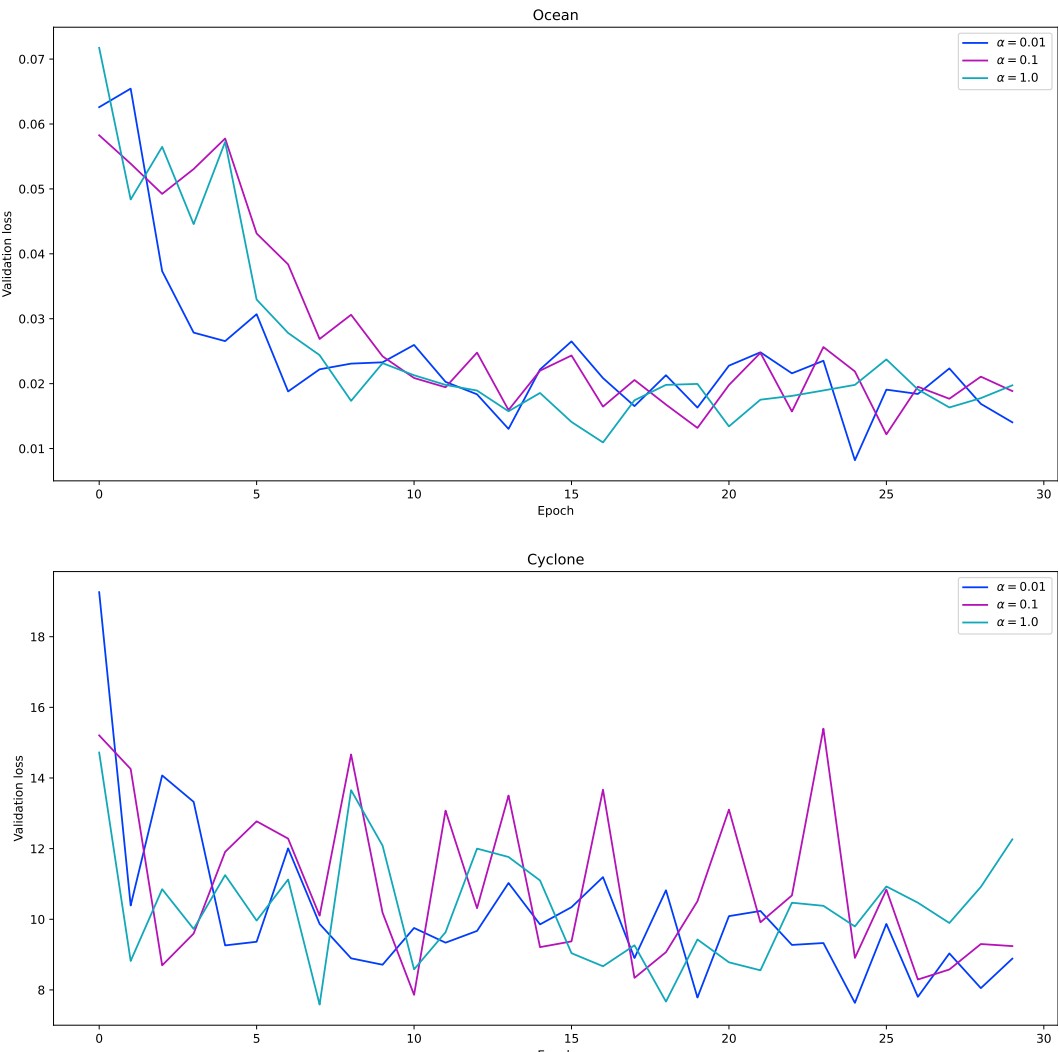

Figure 7: Minimal effect of weight decay in regularising the spiky loss, particularly for cyclone.

# E ABSOLUTE WALL-TIME PLOTS

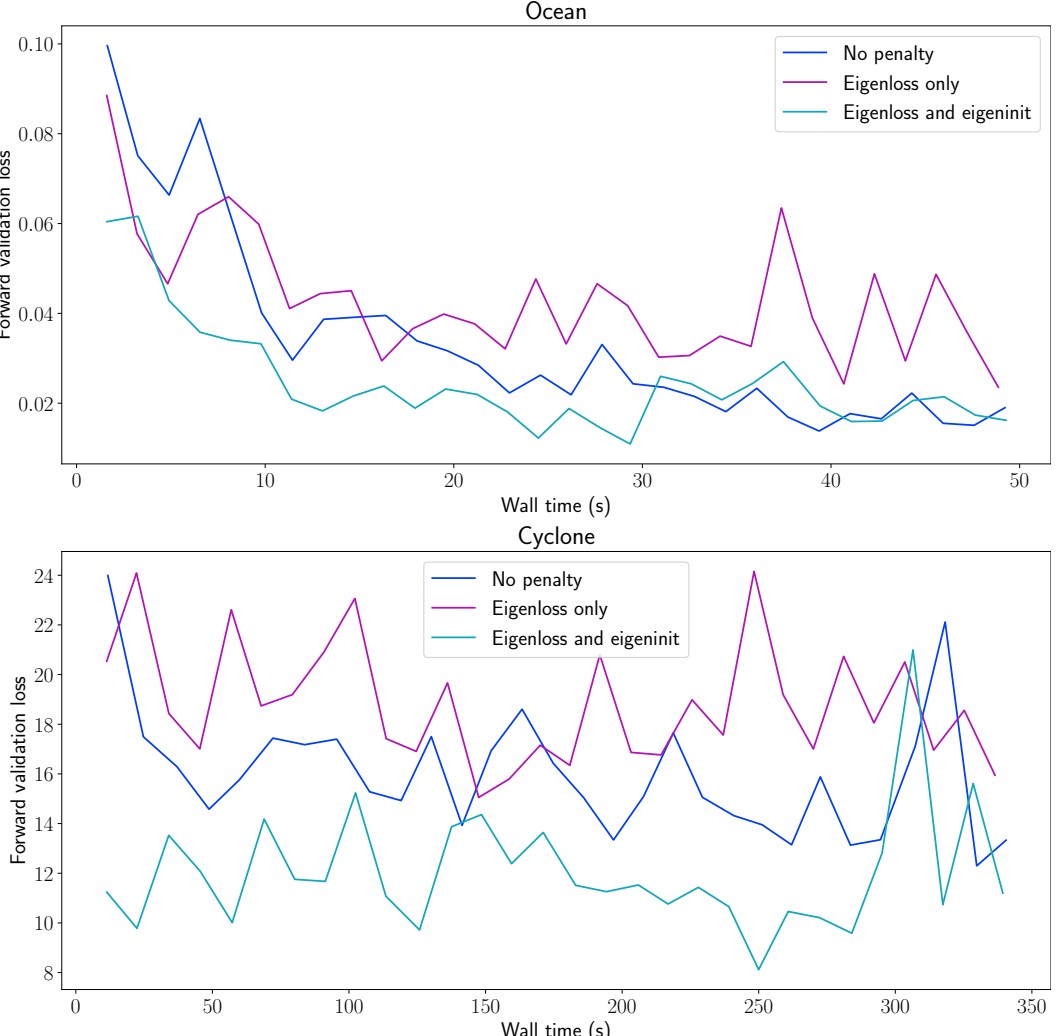

Figure 8: Comparing convergence time on the ocean dataset; this can be seen as a measure of the computational complexity.

