# OpenReview forum: "Eigenvalue Initialisation and Regularisation for Koopman Autoencoders"
_ICLR.cc/2023/Conference — Submitted to ICLR 2023_

### Official Review · Reviewer_AUHg · 2022-10-25

**Confidence:** 3
**Correctness:** 3
**Technical Novelty And Significance:** 3
**Empirical Novelty And Significance:** 3
**Recommendation:** 5

**Clarity, Quality, Novelty And Reproducibility:**

The presentation of the paper is clear, the methods are novel and question is timely. There seems to be enough information for reproducing the results although I did not try to reproduce them myself.

**Strength And Weaknesses:**

**Strengths**
* The problem studied is timely and significant, and is relevant to the ICLR community.
* The introduction and related work sections contain a nice summary of the existing theoretical and empirical results on the initialization and regularization.
* The figures and schematics are clear, helping the reader to follow the arguments. The background on the Koopman Operators (KO) is sufficient and easy to follow.
* Overall, the logical flow of the paper is well-presented. Starting from the intro and background, the problem considered is motivated and connections to the existing literature are mentioned. The presented methods are logical consequences of the existing gaps.
* The presented results specifically on the real datasets help support the arguments.

**Weaknesses**
* I understand that the stable dynamics have a KO with a spectral radius below one, but if the dynamics evolve on a lower-dimensional space some eigenvalues can be closer to zero in magnitude while there are a few eigenvalues with near one magnitude. There seems to be a need to identify the intrinsic dimension of the system in order to find what is the right number of near-one eigenvalues. Did the authors experiment with the effect of model mismatch in this sense? Meaning that if the number of near one eigenvalue considered is not consistent with the intrinsic dimension f the system. And how this affects the overall quality of the results.
* There is no mention of the computational complexity of the system. The way I imagine this to be implemented is either using a parameterization that directly incorporates eigenvalues and the regularizations therein. In this scenario, the errors are backpropagated directly through the parameters (eigenvalues and eigenvectors) but the downside is that in every forward iteration one needs to compute the weight matrix. The second scenario (which is less likely) is if the weight matrix is parameterized in the traditional way, but at every iteration, the eigendecomposition is computed to evaluate the regularization. In both cases, there seems to be some computational burden with jointly initializing or regularizing all the weights. I would like the authors to clarify this (and possibly add a paragraph to the paper) and change the x-axis of the right-hand side plots on Fig. 4 to wall clock time instead of epochs.
* The authors motivate the closeness of the eigenvalue magnitudes to one, but at the bottom of page 5, the distributions considered seem to be a bit problematic. First, doubleGaussianEigen is not a distribution (it should be divided by 2, right?). Second, gaussianEigen and uniformEigen do not seem to encourage the eigenvalues to be of magnitude one. Third, unitPerturb seems to enforce all the eigenvalues to be near one magnitude. Based on my comment regarding intrinsically low-dimensional systems this seems to be a limiting factor for modeling those types of dynamics. Can the authors include a spike and slab type prior and report the results?
* My biggest concern about this paper is with respect to the experiments. Below I elaborate on how the results section can be improved.
    * It is argued that the weight decay regularization is not compatible with the assumptions of stable dynamics. But in the experiments, no comparison against weight decay regularization is presented. Can the authors include the comparisons and ablations (different combinations of initialization (eigeninit, He, Xavier) with regularization (eigenloss, l1, l2)?
    * The results are shown on limited architectures and the effect of the nonlinearity, width, depth, and architectural choices are not investigated systematically.
    * How is the dimension of the KO layer chosen? Can the authors include dimension mismatch studies showing that the method still outperforms alternatives when the wrong dimension is chosen?
    * In Fig. 4, on the right-hand side plots, it seems that all the methods eventually get to similar test errors, if so it's important to compare the time that it takes for different methods to get there. As I suggested above, changing the x-axis to wall clock time will address this. For each method, which epoch is chosen for the results on the left side plot? If the criterion for choosing a  model is the epoch number then the left-side plots and right-side plots seem to deliver the same message. What's more interesting is to change the criterion to convergence. This can address the following question: once all the models achieve similar test performance, which one learns a better dynamics model and generalize better to more time points?

**Minor Comments**
* What does the following line mean and how is this solved by the presented methods?
> Importantly, backpropagation through the eigenvalues of a matrix is numerically stable if there are no repeating eigenvalues.
* Can you include standard (with some sort of normalization that is comparable across different datasets) units on the y-axis of the plots of figure 4?
* Is the following paragraph systematically tested? While it sounds very interesting I would appreciate it if the authors do a systematic experiment where the amount of dissipation changes in the data generating system and show that different values of the hyperparameter are optimal with varying dissipation.
> For instance, if the underlying system is measure-preserving, we can weight $\epsilon_{\lambda}$ more strongly with respect to the reconstruction and prediction losses. In contrast, if the dynamics are dissipative, we can use a weaker weight.
* In the last paragraph of page 3, shouldn't $MSE(x_k, \hat{x_k})$ be changed to $MSE(x_k, \hat{x}_{k+1})$?
* Are the methods only developed for autonomous systems? Can this be extended to input-driven systems (which might be the case for many physical systems)? What would change if one was interested in training a KA with intrinsic dynamics, inputs, and outputs? Do we still need to encourage the eigenvalues to be near one?

## Post Rebuttal

I thank the authors for the clarifications and for taking my comments into account for their revision. Specifically, I appreciate that the authors included the suggested prior, did the wall-clock experiments, included the discussion about the intrinsic dimension of the system, and normalized the y-axis in the results figure.

Given the new plots, I am not quite convinced that the method outperforms alternatives. Fig. 3 which contains the experimental summary of the paper shows that all methods achieve similar validation performances. Given the noisiness of the validation loss plots, it's hard to conclude that the proposed method achieves better results. If the test and validation data have similar statistics the level of noise observed across epochs in the validation loss plots should appear in the test data as well. In addition, the training/validation plots and results of the test data in the table are not accompanied by their corresponding variances. There are multiple ways to include this information, but this is done commonly using random seeds (since the eigenvalues are sampled from spikeAndSlab randomly every initialization will change the eigenvalues which can give rise to training statistics). This is specifically problematic given the computational burden of the eigenLoss raising the question that given a similar computational budget (memory and time) which method achieves the best performance when the within-model variance is accounted for? The experiments included in the supplementary (section E) do not show a clear trend of eigenLoss or eigenInit being helpful for the presented datasets. In the Ocean dataset "No penalty" model performs similarly to "Eigenloss and eigenInit" and in the Cyclone dataset, there's so much noise in the validation loss that it's hard to determine between "No penalty" and "Eigenloss and eigenInit" which one is doing significantly better (also the losses seem very small even in the first time point which is strange).

In the rebuttal round, the authors fixed the conceptual error raised by the other reviewer and re-did all the experiments. The paper is fully restructured and the figures are changed, given that the changes are major, I do believe that the submission requires another round of careful reviewing. Moreover, it's unfair to other submissions since the authors used more time for the major changes. Given this, I tend to keep my score as is. I do believe however that the paper is well motivated, and careful experimentation may better represent the merits of the eigenLoss and eigenInit. Therefore I'd encourage the authors to do another round of polishing and submitting elsewhere if not accepted here.

**Summary Of The Paper:**

The paper investigates the regularization and initialization of the Koopman Autoencoders (KA). KA aims at learning the lifting functions to transform the input dynamical system to the space where the state of the system evolves linearly. Existing methods for well-conditioning the KA rely on specific parameterizations (block-diagonal, tri-diagonal) or indirectly regularize the dynamics (Lyapunov-based, forward-backward soft constraints). Motivated by the fact that KA with stable dynamics has spectral radius with a magnitude less than one, the authors propose to initialize the Koopman layer transition matrix to have eigenvalues near one, and to regularize the model during training to keep its eigenvalues near one. Results are shown on synthetic and real datasets showing that the proposed initialization and regularization scheme outperforms the alternatives on the test data.

**Summary Of The Review:**

The paper proposes a timely solution to an important problem which is well-conditioning the Koopman layer in the Koopman Operators. Overall the paper has good quality, however, the results and experiments can be improved to make the case stronger. I'm happy to increase my score to acceptance if my detailed comments above are addressed.

---

> ### Author Response · Authors · 2022-11-17
> **Comment to AUHg (1)**
>
> ## General Response
>
> We thank AUHg for their detailed comments, and for taking the time to raise further questions. We have addressed the major comments. Further, we have accepted the changes arising from the minor comments. We hope that given our changes and as noted in the review, that you may be open in considering increasing your score to acceptance.
>
> ## Particular Comments
>
> ### Intrinsic dimension of the system
>
> The revised manuscript now includes an ablation study of how model mismatch affects the performance of our techniques, showing that we still outperform previous baselines. Our use of spike and slab initialisation also reduces the effect of model mismatch, as it initialises an appropriate number of unit-circle eigenvalues and allows the rest to handle dissipative aspects of the system (see below). In a related question, we also explore how the number of non-zero eigenvalues in a system may be approximated with dynamic mode decomposition. We hypothesised that we could use this information to inform the prior in the spike and slab distribution i.e. the proportion of unit-circle eigenvalues sampled (see Problematic distributions section below).
>
> ### Computational complexity
>
> We now include plots with wall time on the x-axis instead of just epoch to show that the computational complexity of the eigendecomposition when computing the penalty term does not add any noticeable burden to the backwards pass. We also intuitively justify this by noting that for even our most complex dynamical systems, the size of the Koopman operator approximation used is at most 16x16. Whilst eigendecomposition does scale at roughly O(n^3), as in matrix multiplication, this does not affect the convergence speed, in terms of wall time, of our methods. However, we appreciate the reviewer brought this issue to our attention, as it is certainly something to consider if one required a much larger Koopman approximation.
>
> ### Problematic distributions
>
> We indeed agree that our schemes had several limitations; namely, gaussianEigen and uniformEigen did not necessarily initialise the eigenvalues near 1, whereas unitPerturb only allowed for eigenvalues initialised near 1. To address this, we incorporated your suggestion of a “spike and slab” type prior. This scheme, which initialised a certain proportion of eigenvalues around the unit circle and others with a uniform distribution between 0 and 1, achieved comparable empirical results than even our most recent methods. We have adopted this initialisation scheme into the heart of the paper, as we believe it caters for the intrinsic dimension of dynamical systems which require a certain number of eigenvalues on the unit-circle (corresponding to conserved quantities) and a certain number of eigenvalues inside the unit-circle (corresponding to the dissipative or finite-time horizon dynamics of the system).
>
> However, the use of a spike and slab distribution naturally raises the question of how to choose the parameter \theta which governs the proportion of modulus 1 eigenvalues sampled. Whilst we do so through a simple hyperparameter search, another interesting approach could be to use a dynamic mode decomposition to give an estimate of the number of non-zero eigenvalues and hence the size of \theta. We included this in the revised draft. We compute the DMD of the ocean dataset and examine a sample of n largest eigenvalues, with n being the size of our Koopman approximation. Choosing \theta to be the ratio of unitary eigenvalues to total eigenvalues, we show that choosing the spike and slab distribution in this way gives the optimal initialisation distribution and best test loss. We propose extending this method by examining the eigenvalues of the DMD and determining the dissipative/conservative nature of the system at hand, thus providing a prior for the characteristics of the eigenloss to use in regularisation.

---

> ### Author Response · Authors · 2022-11-17
> **Comment to AUHg (2)**
>
> ### Improvements on Experiments (Addressing Each Dot Point)
>
> *Dot Point 1*
>
> We now include a comparison against different initialisation schemes (eigenInit compared with Xavier) in our main plot (Figure 4), showing that our initialisation outperforms these methods for each dataset. Further, we include ablations in a different section (5.5).
>
> *Dot Point 2*
>
> Respectfully, we believe we have included enough architecture sets for comparison against. In particular, the consistent Koopman autoencoder introduced by Azencot et al. (2020) remains the best performing model for the types of autonomous dynamical systems considered, significantly outperforming other architectures such as Hamiltonian neural networks. In the interest of clarity and space, we believe it is sufficient to compare our methods with a standard Koopman autoencoder and a consistent Koopman autoencoder. Other methods were trialled (i.e., feedforward and Hamiltonian) but performed significantly worse than the above baseline architectures. We also conducted a systematic study of how the width of the autoencoder and the operator affects performance.
>
> *Dot Point 3*
>
> The dimension of the KO layer is chosen empirically based on knowledge of the complexity of the dynamical system at hand. We acknowledge that there may be more rigorous ways to choose the dimension. However, we now include dimension mismatch results showing that even despite varying the dimension out of the appropriate range achieves better results than other methods. Further, our DMD method may be extended to choosing the size of the KO layer, something which we promise to explore in the next few weeks.
>
> *Dot Point 4*
>
> Perhaps most importantly, you noted for Fig. 4 that the left-hand side and right-hand side plots deliver the same message if we use epochs to select the model to plot the cumulative test error with, as all methods eventually get to similar test errors. We wish to make it clear that we use the criterion of minimum validation loss to select the model, and then plot the cumulative test error. This, we believe, addresses the following question: “once all the models achieve similar test performance, which one learns a better dynamics model and generalises better to more time points?”
>
> ### Minor Comments
>
> Our responses to the minor comments are as follows:
> - Regarding the numerical stability of backpropagation through the eigenvalues, we believe our statement is factually correct and common knowledge. In the paper It now reads, "Noteably, backpropagation through the eigenvalues of a matrix is **always** numerically stable, even if there are repeating eigenvalues." If the reviewer requires further clarification, we are happy to discuss.
> - We normalise the y-axis in Figure 4 to make comparison between datasets easier.
> - In the last paragraph of page 3, we have changed the value in MSE to $\hat {x}_{k+1}$
> - Although it will not be finished before the end of the current discussion period, we are conducting a systematic experiment where we vary the amount of dissipation changes in the data generating system and show that different values of the eigenloss hyperparameter are optimal with varying dissipation. We are hoping to combine this with the insights from DMD to provide a rough guide to choosing an eigenloss.
> - As far as input-driven systems are concerned, we have now mentioned this as a direction of potential exploration in our discussion. Due to time restrictions, we will not be exploring it in the current paper.

---

### Official Review · Reviewer_eMgi · 2022-10-26

**Confidence:** 3
**Correctness:** 3
**Technical Novelty And Significance:** 3
**Empirical Novelty And Significance:** 3
**Recommendation:** 5

**Clarity, Quality, Novelty And Reproducibility:**

Question:
- The paper mentions that backprop through eigenvalues is numerically stable if there is no repetition among eigenvalues. However, can we avoid the instability by just using the proposed regularization? If yes, it is not clear to me.
- How are the convergence rate ratios measured? Specifically, Table 3 is not clear what the numbers are and how to compute them. Also, what is "best scheme and control"?


**Strength And Weaknesses:**

### Strength
- The paper is well-motivated in the introduction and method section.
- In my opinion, regularizing eigenvalues is an interesting idea. This is possible for the Koopman operator because of its linearity.

### Weaknesses
- The paper shows an empirical approach, lacking theoretical investigations.This does not mean that I do not appreciate the paper contributions.
- Even as an empirical study, the experiments of the paper fall short and are not comprehensive enough. For example, the paper may need to ablate the effect of changing the weight of $\epsilon_\lambda$. It’s not clear what the initialization strategy (among 4), and the regularizer (among 2) are used in the main text (Table 1, 2 and Figures 4).


**Summary Of The Paper:**

The paper presents an initialization strategy for Koopman operators and a loss regularization in terms of spectral information for Koopman autoencoders. For initialisations, the paper claims that approximated Koopman operators should be initialized such that their spectral norm should be less than 1. For regularization, the loss is penalized such that the eigenvalues are concentrated on the unit circle.

**Summary Of The Review:**

In general, the paper proposes an nteresting way for egeinvalue intialization and regularization for Koopman operators in dynamic systems. However, the experiments should be polished more to support the claims.

---

> ### Author Response · Authors · 2022-11-17
> **Comment to eMgi**
>
> ## General Response
>
> We thank eMgi for their comments, and are delighted they found the idea interesting. We also appreciate that they found the paper well-motivated.
>
> ## Particular Comments
>
> ### Weaknesses
>
> Regarding weaknesses, we have attempted to strengthen the empirical approach by moving to a “spike and slab” distribution for initialisation (see also our response to AUHg’s comments). We also choose the hyperparameter $\theta$ in the spike and slab distribution by estimating the number of unit-circle eigenvalues of the Koopman operator using dynamic mode decomposition (DMD). We believe this is a unique approach and strengthens the intuition with which we proposed eigenvalue initialisation. The DMD approach may also estimate the conservative/dissipative nature of the system and thus inform the appropriate eigenloss to use. Finally, we show that this method DMD seems to predict the optimal value of \theta for spike and slab, leading to the lowest overall test loss on the ocean dataset over several runs.
>
> Additionally, we have performed extensive ablation studies to show how varying the numerous hyperparameters impacts the effectiveness of our methods (section 5.5). Since this seemed to be the main weakness suggested by this reviewer, we hope that the additional result provides evidence of our performance across a range of hyperparameters and demonstrates empirically the effect of eigenloss and eigeninit across the various types of datasets.
>
> ### Clarity Questions
>
> In addition to polishing the experiments, we have made minor changes to improve the clarity of things such as the convergence definition which can be seen in Appendix C, as well as other terms which were not as clearly defined in the previous draft.
>
> Concerning the stability of the eigenvalues, the sentence has been corrected and now reads as follows: "Noteably, backpropagation through the eigenvalues of a matrix is **always** numerically stable, even if there are repeating eigenvalues."
>
> ## Conclusion
>
> We consider our changes in response to eMgi's comments to have really strengthened the experiments and overall position of the paper. Considering they seemed to have similar comments to AUHg, we hope that eMgi may consider increasing the score to accept. We are also happy to answer further questions and tighten experiments further if eMgi would like to see additional results.

---

### Official Review · Reviewer_ibgi · 2022-10-27

**Confidence:** 4
**Correctness:** 2
**Technical Novelty And Significance:** 3
**Empirical Novelty And Significance:** 2
**Recommendation:** 3

**Clarity, Quality, Novelty And Reproducibility:**

**Clarity**

I think it would be good to elaborate more on when the "spectral radius is <= 1" assumption and the eigenloss are helpful and when they aren't applicable.

- I understand that it **can** be advantageous to require/encourage that the spectral radius is <= 1 for stability. However, this might be restrictive. For example, in the fluid flow example in the Lusch et al., 2018 paper, trajectories that start inside the attractor exhibit growth until they reach the attractor. I know this sentence is motivation to focus on systems with spectral radius is <= 1: "Namely, Koopman eigenvalues of stable dynamical systems are constrained in the unit circle of the complex plane (Mauroy & Mezic, 2016)." However, I could be wrong, but I think this paper has a more narrow scope than this statement. From the abstract: "The main results establish the (necessary and sufficient) relationship between the existence of specific eigenfunctions of the Koopman operator and the global stability property of hyperbolic fixed points and limit cycles." The paper seems to focus on hyperbolic attractors and not systems with, say, a non-empty continuous spectrum.
- Further, it seems that the eigenloss encouraging eigenvalues to have magnitude near 1 could be detrimental for systems that have decay toward an attractor or dissipation, as mentioned in the paper. It's mentioned that in the case of dissipation, there could be a weaker weight. However, would you know that you have dissipation and should use a weaker weight? And for significant dissipation, would you be better off with no eigenloss in that case? For example, a common test case for Koopman methods is the "2D fixed point attractor" in the Pan & Duraisamy (2020) paper, which has two eigenvalues, both causing decay. I suspect that the eigenloss would be detrimental for this example.
- Again, I agree that these assumptions **can** be helpful, so I think this method has value, but I think it's useful to make it more clear that these assumptions aren't always helpful.

I spent a while trying to understand which combinations of ideas were tested, so I think the language in Section 5 could be more clear. For example, in Table 1, I'm guessing that "None" refers to the Standard KAE, and "init," "loss," and "both" refer to models with the eigninit and/or eigenloss ideas applied but not the "consistency" idea? And then "consistency & best" refers to using the consistency idea plus the best of the previous 3 columns?

Relatedly, I don't understand the sentence "The particular schemes applied to the consistency architecture were chosen by a two-stage selection process: first employing the best dataset eigenloss or eigeninit and then taking the lead performer out of these."

How is cumulative test error defined? Averaging test error over the prediction steps?

I'm confused about Figure 5. The left side and the right side have very different sizes of eigenvalues. Are they both accurate? It doesn't make intuitive sense to me that the same dynamics could be predicted well with such different sets of eigenvalues.

**Quality/Correctness**

"All models with both eigeninit and eigenloss have significantly lower final validation loss than the control." I think this is an unfair summary of the table since test data is what ultimately matters, not validation data.

There are a couple of mathematical details that are a bit incorrect.
- The discussion around "eigenvector diminishes with time... the norm of v_j does not change..." is not quite right. Typically, eigenvectors are normalized. As the exponents on the eigenvalues increase, that does not change the eigenvectors.
- "The Koopman operator is a linear object with a complex spectrum, i.e., the associated eigenvalues are complex-valued, when eigendecomposition exists." The Koopman eigenvalues are not necessarily complex. For example, the "2D fixed point attractor" in the Pan & Duraisamy (2020) paper has real eigenvalues only. Of course, you could say that real numbers are a special case of complex numbers, but I still think that the sentence is misleading because then it's an empty statement: of course the eigenvalues are complex when you count real numbers because what else could they be? The next sentence is also a bit wrong: the Lusch et al. (2018) paper mixes blocks for complex conjugate eigenvalues & blocks for real eigenvalues.

In Table 1, first row, the consistency result should be bolded as best, but the init result is bolded.

**Reproducibility**

- There are several options for eigenvalue distributions on the bottom of page 5 and two options for the eigenloss, but then it's unclear which one(s) work the best and with which values of parameters such as sigma. There is some hint of this in the appendix, but not a complete answer. It seems that the best option is inconsistent.
- The "Standard KAE" is not clearly defined. I'm guessing it's the Azencot et al. (2020) model with some consistency features removed? Specifically adding the backward & consistency losses from Azencot et al. (2020)?


**Strength And Weaknesses:**

**Strengths:**

I think it makes sense that if you have some physical knowledge of the system, you could use that to influence the eigenvalues of the Koopman matrix and have better results. I appreciate that this was tested on four datasets, which is a pretty good number. The improvements over the Standard KAE are pretty strong.

**Weaknesses:**

My primary concern is correctness: When the initial Koopman matrix is defined with new eigenvalues, it may be complex. The real part of each matrix element of A is kept. Does that change the eigenvalues, or are they still \lambda^{\tilde}? I tried a small test, and it seems that the eigenvalues change. Please let me know if I'm missing something. I set A = [1, 2; 3, 4]. The eigenvalues are approximately  -0.3723 and 5.3723. I changed the second eigenvalue to 0.5 + .1i and reconstructed the matrix: A_new = V D_new inv(V). The eigenvalues are as expected, but the matrix is complex. I took the real part: real(A_new) and the eigenvalues change to approximately -0.3723, 0.5.

It would be nice to have more benchmark methods to compare against. In lieu of that, I think some of the language should be softened. I think it's overstatement to say that the results are compared against "several state-of-the-art baselines." Similarly "was introduced by Azencot et al. (2020), yielding stable Koopman systems and state of the art performance" is an overstatement, since that paper only compares to one of many previous papers. (There are many papers with Koopman autoencoders cited in this paper.) Further, the Azencot et al. (2020) comparison to the Lusch et al., 2018 paper is not a complete one, since they left off the key feature of letting eigenvalues vary for a continuous spectrum.

It's mentioned that "backpropagation through the eigenvalues of a matrix is numerically stable if there are no repeating eigenvalues." It sounds expensive to backpropagate through an eigendecomposition. How fast is this method? Does converging in fewer epochs come at a cost of each epoch being much slower?

Although this paper shows improved results with the eigenloss and/or eigeninit approaches, there's a lot of a variability in whether it's best to use one or the other or both, and the improvements are much more modest when compared to the consistency baseline. The only column that is better than the consistency baseline more than half of the time in terms of test data the init-only column. Looking at Table 1 & Figure 4, if you were to try all of these approaches for a new dataset and pick the best approach based on the validation loss, that would usually not correspond to the best approach based on the cumulative test error. Is this because of differences in loss vs. error, or are there variable amounts of overfitting?

I also have some comments in the next section.

**Summary Of The Paper:**

This paper is about improving Koopman autoencoder models in two ways: (1) a better initialization of the Koopman matrix by directly changing the eigenvalues, and (2) penalizing the eigenvalues of the Koopman matrix during training. These changes are tested on 4 datasets (some synthetic & some real-world), showing improved convergence and long-term prediction error.

**Summary Of The Review:**

I think that this paper presents a couple of interesting ideas and shows results that are sometimes better than the Azencot et al. (2020) paper. However, I think that the eigeninit is not doing what they think it is: it seems that the eigenvalues will change when they take the real part of the new matrix. I believe there are multiple incorrect or exaggerated statements. Also in order to improve upon the Azencot et al. (2020) paper, many models were trained with variants of the new ideas, so (a) the improvements may be mostly due to having many models to pick from, and (b) since there aren't consistent winners, there may need to be a lot of tuning to use these ideas.

---

> ### Author Response · Authors · 2022-11-17
> **Comment to ibgi (1)**
>
> ## General Response
>
> We really appreciate the detail of the reviewer’s comments. We agree with almost all of them and have tried to take as many steps as possible to address them in the limited time between their publication and revision upload. We hope that the new sections we have introduced, and the correction of your primary concern might allow you to reconsider your position on the paper. We believe that a first step toward regularisation strategies in Koopman methods will be useful to the field even if they are not as widely useful as something like weight decay. We would point to the success of our schemes on the real-world datasets as evidence for this applicability.
>
> ## Particular Comments
>
> ### Primary Concern of Eigenvalue Initialisation
>
> Our claims concerning the original eigenvalue initialisation were indeed flawed. The real transformation indeed alters the eigenvalue in a meaningful way. Consequently, the distributions we were aiming for, were not necessarily reached. As such, we reformulated how we initialise the eigenvalues, which is explained in a revised version of section 4.1. In the reformulation we only alter the magnitude of the eigenvalues leaving the phases intack. We also take care to alter the magnitude of complex conjugate eigenvalue pairs in the same way. Doing so, if the original matrix U was real then the reconstructed matrix \tilde{U} is also real (up to machine presition) and therefore.
>
> We reran all relevant experiments using the reformulation of the eigenvalue initialisation, the results of which can be seen in sections 5.2-5.6.
>
> Having realised our mistake, we still wondered why the previous methods worked. As such we ran tests and observed that they still represented an increase in the average eigenvalue magnitude when compared to Gaussian or Xavier initialisation. This is what likely made them successful in our eyes and supports our hypothesis that higher eigenvalue magnitudes are needed.
>
> Again, we thank the reviewer for their careful read of our original manuscript and the detailed comments they provided.
>
> ### Benchmark Methods
>
> We believe more benchmark methods would be productive to include. However, we are unaware of any architectures that do outperform the consistency benchmark on our datasets. If you do have one in mind, we would be very happy to run an experiment augmenting it with our schemes. In addition, we did run benchmarks against traditional deep learning architectures like predictive autoencoders and feed forward networks for the old initialisation schemes. However, these performed worse than our methods. We did not include these comparisons in the appendix since they employed the old methodology. However, we are happy to report them in the discussion period.
>
> We would also like to address concerns regarding the non-existence of continuous spectrum benchmarks. The fluids benchmark does have a continuous spectrum and it is likely that the cyclone system also has a continuous spectrum. Although we do not admit a continuous spectrum for our operator this shows we still have utility in that domain. We are willing to introduce an architecture which could accommodate these problems in a new revision of the paper if we are accepted.
>
> Concerning the remark regarding language: we have softened the language and our claims throughout the manuscript.
>
> ### Numerical Stability and Efficiency of Eigendecomposition
>
> Here, we do agree that discussion of computational complexity of the eigendecomposition was a meaningful omission in the original paper. Although we had run similar experiments for personal confirmation, we were not explicit about the computational cost. We address this with the new section 5.5 where we run wall time for the largest operator sized architectures used for ocean and cyclone. Although eigendecomposition scales cubically, note that the Koopman operator acts on the reduced state space and, therefore, the operators we tried do not get any larger than 16x16 and, even if they do, various elements of GPU computation actually make eigenvalue calculation quite efficient at larger scales.
>
> Concerning numerical stability of the eigenvalues, we have rephrased the sentence to say, “Importantly, backpropagation through the eigenvalues of a matrix is always numerically stable, even if there are repeating eigenvalues.”

---

> ### Author Response · Authors · 2022-11-17
> **Comment to ibgi (2)**
>
> ### Variability in choice of scheme and inconsistent validation loss
>
> As mentioned in the general statement, we have sought to address the issue of choice by limiting our library members to unit_circle_mse for the eigenvalue loss and to spikeAndSlab for the eigenvalue initialisation. Consequently, one must simply optimise the hyperparameters of these. To tune the hyperparameters one can use common techniques provided in the literature for an analogous protocol like weight decay. In addition, tuning θ via the dynamic mode decomposition (DMD) method provided does have direct correlation between test performance and validation performance. Similarly, the ablation shows common values for ε which work to improve all datasets.
>
> ### Elaborating on choice of systems with limited spectral radii
>
> We agree that requiring a spectral radius smaller than 1 can be restrictive and that we do largely focus on systems which we know are in this regime. We have two comments to address this point.
>
> Firstly, our proposed schemes can easily be adapted to other radius constraints, for example to promote a certain number of eigenvalues outside of the unit circle if required.  One just needs to change to mean and variance of the uniform/spike initialisation or the functional form of the loss term.
>
> Secondly, we argue that the cyclone dynamical system is not necessarily within the class of dynamical systems where \rho(U)<=1 and that our sch  emes still perform well there. Indeed, it is not measure-preserving and due to convection would include some chaotic phenomena. As such, the consistency architecture performs poorly whereas our schemes help with convergence and test error. One of the reasons we only create a soft bias on the eigenvalues is to account for this very case. We believe that the fact that we are not as restrictive as other papers is one of our strengths.
>
> ### The weight of eigenloss
>
> Concerning the weight of unit_circle_mse, it is certainly true that it could be detrimental for systems with heavy dissipation such as the fixed-point attractor. We have three comments to address this.
> - Firstly, we hope that our new section on DMD-inspired hyperparameter choice and various ablation could provide an initial guess for the weighting.
> - Secondly, we believe our initialisation would still be key in such a system. Even in this case, a gaussianElement initialisation is still not optimal since there is one eigenvalue significantly larger than 0.
> - Thirdly, we do not seek to provide a scheme which will necessarily help in all cases. Eigeninit will probably be more general than eigneloss and may not work in every case. Like standard regularisation and architecture choices in practice, the best option will depend on the dataset and task at hand.  We note however that the proposed schemes work well across all datasets considered .
>
> ### Understanding which combinations were used
>
> Since we have limited the number of candidates to one for each scheme this should be a lot clearer. The hyperparameter values are provided in Appendix A.
>
> ### Two-stage selection process
>
> Again as above, we have removed the other schemes.
>
> ### Cumulative test error
>
> The cumulative test error is the summed average test error up to that point in time.
>
> ### Confusion about figure 5
>
> We have double checked the figure; the figure is accurate. We do not believe it is intuitive either and is the reason we introduced our schemes. It appears the network is unproductively compressing the observables into a state where dissipative dynamics can still provide high accuracy. This observation is one of the reasons why we introduced these new schemes.
>
> ### Unfair summary of results table
>
> We have removed this language.
>
> ### Discussion around eigenvalue diminishing
>
> We have altered the language here to refer instead to the action of the eigenvector.
>
> ### Complex Spectrum of Linear Operator
>
> We agree with the assessment of this statement. We have removed the discussion of the eigendecomposition and simply state that the operator has a complex spectrum.
>
> ### Reproducibility
>
> We have tried to address concerns about the best option by stating hyperparameter values and giving a more limited set of initialisation and regularisation options. In addition, we have defined the standard Koopman autoencoder in the text. Indeed, what we propose is the “Azencot model” with consistency features removed.
>
> Full code and scripts to run the baselines and experiments will be released upon acceptance, as noted in the reproducibility statement.

---

### Author Response · Authors · 2022-11-17
**General Statment to All Reviewers**

We would like to provide a general response to the reviews before addressing each individually. First, we would like to thank all the reviewers for the detail they provided in their responses and for their kind words. Because of the detail and constructive nature of the feedback, many elements of the paper have changed, and we are committed to continue these positive changes if we are accepted. Indeed, we believe that current alterations merit a serious reconsideration of recommendations.  We ask that reviewers take time to complete this reassessment and to ask as many questions as possible – we are open to running many additional experiments and reporting their results in the discussion period.

We took the primary concerns with the paper to be the correctness of the initialisation scheme, experimentation with hyperparameters (size of the operator, the weighting of eigenvalue penalties and the type of distribution), ablation with other well-tested schemes such as Xavier initialisation and weight decay, and finally, the computational complexity of our methods.

Ibgi was indeed correct when they pointed out how our original method of reconstruction of the Koopman matrix altered the eigenvalues with the real transformation. We fixed this concern by taking a different approach to initialisation which retained the phase and conjugacy of the original eigenvalues and only alters their magnitude. Having done so, we reran _all_ experiments with the new version and achieved comparable performance. Please see our specific comment for more details.

We agree that hyperparameter experimentation and the clarity of the best scheme was lacking in the original paper. To remedy the scheme clarity, we limited our eigenloss to unit_circle_mse which was the best performing candidate in the original paper. We also limited the number of initialisation schemes: a new candidate suggested by AUHg, spikeAndSlab. To explore the hyperparameter space of these candidates we created a new section (5.5). In this section, we tested against different penalty weights for eigenloss, the operator size and the width of the network. We also altered the hyperparameter governing spikeAndSlab.

To address concerns around the performance of Xavier and weight decay, we included comparisons with Xavier in figure 4 and weight decay in a new section of the appendix (Appendix E).

Finally, we discussed the computational complexity of our schemes in a new section of the results (sec 5.4). Unfortunately, we did not have the time to implement wall time for all our datasets so figure 4 could not be changed in this way. Instead, we made an argument concerning the complexity of eigenvalue decomposition, and provided will time tests for the ocean and cyclone datasets which can be seen in Appendix F. These tests showed that the calculation of the eigendecomposition seems insignificant for the largest operator sizes used and that our schemes still improve convergence speed.

Given the opportunity, we would be happy to improve our submission based on the reviewers’ comments, within the review process of ICLR.

---

### Decision · Program_Chairs · 2023-01-20

**Decision:**

Reject

**Justification For Why Not Higher Score:**

As mentioned in the summary above: this is a good paper, but requires fairly extensive revisions, particularly related to the (revised) method's soundness as well as a proper treatment of computational complexity. I believe this set of revisions is beyond what is feasible at this stage to merit an accept decision.

**Justification For Why Not Lower Score:**

N/A

**Metareview: Summary, Strengths And Weaknesses:**

The paper studies the problem of training Koopman Autoencoders, which is a family of models used to solve physics-informed learning problems. For the Koopman layer in such models, the paper proposes a special type of eigenvalue-tailored initialization (as compared to standard random init schemes such as He or Xavier). Moreover, the model is regularized to ensure that the eigenvalues are kept close to unity. Experiments on both synthetic and real datasets indicate the benefits of this approach.

On the positive side, the reviewers agreed that this is a well-motivated paper that proposes an intuitive approach and validates it on real datasets.

On the negative side, several concerns were raised by the reviewers. Several claims about the method were not well-supported (or over-stated); detailed ablation studies were lacking; comparisons to other standard regularization procedures (such as weight decay) were missing; and the numerical stability of eigenvalue computation was a challenge. A big concern (echoed by multiple reviewers) was computational complexity: eigenvalue computation is slow, and it is not quite clear how the approach scales up.

In the response phase, the authors did address many of these concerns by providing a revised version of the paper. However, the issue of computational complexity (and wall-clock running time comparisons) still persists, particularly on larger problem instances. The issue of real/complex eigenvalues also persists: the authors dealt with this by using a "hack" (in my opinion) and only adjusting eigenvalue magnitudes, but this somewhat weakens the principled motivation behind the proposed method.

The authors are encouraged to consider the above comments while preparing a new revision.

**Summary Of Ac-Reviewer Meeting:**

N/A